# A Decade’s Change in Vegetation Productivity and Its Response to Climate Change over Northeast China

**DOI:** 10.3390/plants10050821

**Published:** 2021-04-21

**Authors:** Min Yan, Mei Xue, Li Zhang, Xin Tian, Bowei Chen, Yuqi Dong

**Affiliations:** 1Kay Laboratory of Digital Earth Science, Aerospace Information Research Institute, Chinese Academy of Sciences, Beijing 100094, China; yanmin@aircas.ac.cn (M.Y.); zhangli@aircas.ac.cn (L.Z.); chenbw@aircas.ac.cn (B.C.); dong_001207@163.com (Y.D.); 2Patent Examination Cooperation (Beijing) Center of the Patent Office, CNIPA, Beijing 100160, China; shemayxm@hotmail.com; 3Institute of Forest Resource information Techniques, Chinese Academy of Forestry, Beijing 100091, China

**Keywords:** vegetation net primary productivity, BEPS, partial correlation analysis, lag analysis

## Abstract

In this study, we simulated vegetation net primary productivity (NPP) using the boreal ecosystem productivity simulator (BEPS) between 2003 and 2012 over Northeast China, a region that is significantly affected by climate change. The NPP was then validated against the measurements that were calculated from tree ring data, with a determination coefficient (*R*^2^) = 0.84 and the root mean square error (*RMSE*) = 42.73 gC/m^2^·a. Overall, the NPP showed an increasing trend over Northeast China, with the average rate being 4.48 gC/m^2^·a. Subsequently, partial correlation and lag analysis were conducted between the NPP and climatic factors. The partial correlation analysis suggested that temperature was the predominant factor that accounted for changes in the forest NPP. Solar radiation was the main factor that affected the forest NPP, and the grass NPP was the most closely associated with precipitation. The relative humidity substantially affected the annual variability of the shrub and crop NPPs. The lag time of the NPP related to precipitation increased with the vegetation growth, and it was found that the lag period of the forest was longer than that of grass and crops, whereas the cumulative lag month of the forest was shorter. This comprehensive analysis of the response of the vegetation NPP to climate change can provide scientific references for the managing departments that oversee relevant resources.

## 1. Introduction

Carbon dioxide (CO_2_) in the atmosphere has increased from 277 ppm in 1750 to 405 ppm in 2017 [1], which has mainly been ascribed to the combustion of fossil fuels, deforestation, and changes in land use. The terrestrial ecosystem is an important carbon sink and source, storing large amounts of carbon in plant parts and the soil [2]. Accordingly, the accurate assessment of the carbon distribution among terrestrial vegetation is important for properly understanding the global carbon budget, supporting climate policies, and undertaking project vegetation management in the future. Net primary productivity (NPP) is one of the components of the carbon cycle and represents vegetation biomass increases after accounting for autotrophic respiration [3,4,5,6].

The ecological process models, which rely on ground data, including climatology, meteorology, vegetation conditions, and soil conditions, are crucial methods for depicting vegetation activities (e.g., carbon, nitrogen, and water cycles) [7,8,9]. The manifestations of ecological processes are complex and diverse, including the physiological and ecological mechanisms of the plants themselves, the community succession of the ecosystem, and the numerous vegetation disturbances in the ecosystem. The boreal ecosystem productivity simulator (BEPS) is a process mechanism model that is driven by remote sensing data, and has been widely used in simulating vegetation productivity and evapotranspiration [10,11,12]. Through the endeavors of many scholars, the original BEPS has experienced numerical developments. A “bucket” module was replaced by a three-layer soil module (0.10, 0.25, and 0.80 cm depth) [13]. Due to the simplified soil moisture mode that ignores the horizontal exchange of water, further processes, such as surface runoff, are added to the water circulation to improve the simulation accuracy of vegetation productivity [14]. After a series of model developments, BEPS is universally applied for depicting many important ecological processes and the interactions between these processes and the surroundings.

Northeast China is covered by rich vegetation types (forest, grass, cropland) and is one of the regions that are most sensitive to climate change [15], which is expected to affect forest growth, productivity, mortality, distribution, and biodiversity [16]. Specifically, the rise in CO_2_ concentration may have impacts on the vegetation NPP through modifying photosynthesis [17]. Therefore, quantifying the vegetation NPP and the relationship between the NPP and climate change is necessary for the carbon cycle and vegetation management. Currently, several previous studies have investigated the NPP and climate change over Northeast China. For example, Zhu et al. [18] estimated the monthly NPP of the Northeast China Transect from 1982 to 2000 and analyzed the response of the NPP to climate change. Peng et al. [15] quantified the response of forest carbon to future climate change in Northeast China. Both studies suggested that climate change will increase the vegetation NPP in future decades. However, these studies were limited to the validation of regional NPP and were focused on the simultaneous interactions between meteorological data and vegetation indicators. The dynamics of the NPP are substantially linked to climatic variations [19]. Additionally, the time-lag effects of vegetation NPP responses to climatic factors should be considered for the comprehensive analysis over Northeast China [20,21].

Therefore, we proposed this study to (1) model the vegetation NPP over Northeast China using the BEPS model, (2) understand the spatial and temporal patterns of the NPP from 2003 to 2012, and (3) investigate the response of the NPP to climatic factors (temperature, precipitation, radiation, and humidity).

## 2. Results

### 2.1. Validation of the BEPS Model

Figure 1 shows a comparison between the annual average NPP for 2003–2012 that was obtained from the BEPS model and that from observations at forest sites over Northeast China. Generally, the BEPS model performed well at simulating the annual NPP, with *R*^2^ = 0.84, *RMSE* = 42.73 gC/m^2^·a, and *p* < 0.01. The mean bias of the simulated NPP was 55.82 gC/m^2^·a, indicating that the BEPS model slightly overestimated the NPP at forest sites. The scale mismatch between the forest plots and the resolution of the input data could be a major reason for the discrepancy. However, the overall satisfactory performance of *R*^2^ and *RMSE* suggested the potential of the BEPS model regarding simulating the regional NPP and analyzing the NPP patterns.

### 2.2. NPP Spatial Patterns and Temporal Trends

The spatial pattern of the annual average NPP over Northeast China is shown in Figure 2a. The decreasing gradients from northeast to southwest of the simulated NPP were significant. The annual average NPP was generally higher than 400 gC/m^2^·a in Daxing’anling, Xiaoxing’anling, and the Changbai Mountains, where natural forests (e.g., evergreen forests) predominated. The annual average NPP ranged from 200 to 400 gC/m^2^·a over Songnen, Liaohe, and Sanjiang plains and the agricultural area at the junction of the Hulunbeier Grassland and Daxing’anling, as well as the burned forest area in Daxing’anling. The annual average NPP was usually lower than 200 gC/m^2^·a in Hulunbeier, west Songnen Plain, and the Horqin Sandland, with the land surfaces covered by grass, crop, and sand.

The relationship between the annual NPP and the time for each pixel was established to calculate the NPP trend (Figure 2b). The results indicated that the NPP experienced a remarkable increase (*p* < 0.05) at the average rate of 4.48 gC/m^2^·a. The NPP with an increasing tendency accounted for 65% of the total NPP, and mainly occurred in forests over Daxing’anling, Xiaoxing’anling, and Changbai Mountains. The NPP showed a decreasing trend that was mainly concentrated on grass, crop, and disturbed forest areas.

### 2.3. Partial Correlation Analysis between the NPP and Climatic Factors

The partial correlation of the NPP with temperature, precipitation, solar radiation, and relative humidity during the growing seasons of 2003 and 2012 was analyzed. The temperature over Northeast China is extremely low in winter and vegetation almost stops growing. As the vegetation growth in this period has no linear relationship with temperature, only the relationship between the NPP in the growing season and the temperature was investigated. Over 56% of the NPP was positively correlated with temperature, with Daxing’anling, Xiaoxing’anling, and the Sanjiang Plain being the most obvious. Correlation between the NPP and precipitation over Northeast China indicated spatial variations (Figure 3b). The crop NPP over the Songnen and Liaohe plains correlated positively with precipitation; the other crop NPP had no significant correlation with precipitation caused by irrigation and snow melting. Most of the grass NPP demonstrated a positive correlation with precipitation over Hulunbeier Grassland and Horqin Sandland, where low temperature and high solar radiation appeared frequently. The forest NPP, especially in Daxing’anling and Xiaoxing’anling, was negatively correlated with precipitation.

The positive effect of solar radiation on the vegetation NPP was mainly reflected in the central and western parts of Daxing’anling and part of the Changbai Mountain, with a correlation factor over 0.6. However, the grass NPP that was distributed in the areas with a low elevation showed a negative correlation with solar radiation. Relative humidity reflects the moisture content in the air and is substantially affected by precipitation and solar radiation. Relative humidity influences the opening and closing of plant stomata, which, in turn, affect the physiological processes of vegetation, such as transpiration and photosynthesis. Relative humidity that is too high or too low will cause the plant stomata to close, slowing down or even stopping photosynthesis. Overall, relative humidity was positively correlated with the grass and crop NPPs. Since vegetation transpiration was negatively correlated with relative humidity, high relative humidity preserved more water for vegetation growth. Relative humidity was negatively correlated with the forest NPP because the forest accumulated water through its developed root system.

### 2.4. Lag Response of the NPP to Precipitation

Figure 4 shows the results of the lag analysis between the NPP and precipitation in Northeast China, with the first column showing the maximum correlation coefficients between precipitation and the NPP. The areas with a maximum correlation coefficient greater than 0.3 (*p* < 0.05) were defined as areas where the NPP was correlated significantly with precipitation. In May, the NPP showed a significant correlation with precipitation, but only accounting for a tiny proportion over Liaohe Plain, Daxing’anling, and Songnen Plain. Between June and August, the significant correlation between NPP and precipitation mainly occurred in the eastern part of Hulunbeier, Chifeng City, and the agropastoral areas of the Songnen, Liaohe, and Sanjiang plains, as well as the southwestern part of Daxing’anling and the middle part of Xiaoxing’anling. The proportion of the significant correlation between NPP and precipitation decreased in September, which was predominant in the grassland in the northwest of the study area.

The spatial distribution of the lag periods in the response of the NPP to precipitation is shown in the second column of Figure 5. The lag period in the response of the NPP to precipitation increased with time. During July and August, the lag periods of most of the forested areas were 2 to 3 months, which was generally longer than those of the grass and crop areas.

The third column in Figure 4 shows the spatial distribution of the cumulative lag months in the response of the NPP to monthly precipitation. The percentage of one cumulative lag month was over 50% in May, whereas the proportions of other cumulative months were small. The NPP became more significantly correlated with the precipitation of the current month during June and July, when the proportion with five cumulative lag months dominated due to the abundant precipitation. The results manifested a decreased correlation of the NPP with the monthly precipitation, but an increased correlation with the cumulative precipitation of several months. Lastly, the distribution of the cumulative lag months of the vegetation NPP was even in August. Owing to decreased precipitation at the end of the growing season, vegetation growth relies more heavily on soil moisture and water storage in the roots. Therefore, the lag in the response to the cumulative precipitation of four or five months was obvious.

## 3. Discussion

### 3.1. Improvement of the Regional Vegetation NPP Simulation

The regional NPP was quantified over Northeast China using the BEPS model in this study, where the model’s input data were improved using GLASS *LAI*. As a key variable in the model, *LAI* directly reflects the carbon content of vegetation leaves, even the vegetation growth and development. Currently, GLASS *LAI* products have been validated well against various sites around the world. However, overestimation remained for regional forest NPP during the validation against tree ring data, indicating that uncertainties caused by that were model parameters require further optimization. Some parameters in the biological functions of the BEPS model usually varied significantly at seasonal and interannual scales; for example, the foliage clumping index (*C_f_*), the slope of stomatal conductance to the net photosynthetic rate (m), the maximum photosynthetic carboxylation rate (*V_cmax_*), and the electron transport rate (*J_max_*). Therefore, sensitivity analysis is necessary, which calculates the main contributors among the parameters and provides potential prerequisites for model parameter optimization [22,23]. Data assimilation has also shown itself to be a potential technique for adjusting model parameters and updating model state variables, and even improving modeling accuracy [24]. This strategy considers the errors of input data, parameters, and model structure, providing an effective scheme for carbon and water fluxes.

As vegetation productivity was simulated based on a 1 km scale, a problem of mixed pixels occurred, which is a common problem within this study area due to the heterogeneity of regional landscapes. Accommodating the decomposition of mixed pixels using high-resolution remote sensing data is a crucial issue for the process-based model. Additionally, the high-resolution remote sensing data could improve the accuracy of the land classification map and provide more specific vegetation information, which are important input data for the BEPS model.

### 3.2. Response of the Vegetation NPP to Climate Change

Temperature is the major limiting factor for vegetation growth, especially in the cold temperate zone [25], which explains the weakening partial correlation between NPP and temperature from the north to the south of the study area. Similarly, precipitation is also the leading factor for vegetation growth; however, most of the forest NPP was negatively correlated with precipitation over Northeast China. This was caused by several factors: First, the underlying surfaces of the forest were rich and presented a certain water storage capacity. Second, as the BEPS mode splits the soil into three layers for the simulation of water processes and as the forest is characterized by a developed root system and high water storage capacity, there was a lag in the influence of precipitation on the forest NPP. High relative humidity will lead to the closure of vegetation stomata, causing a reduction in CO_2_ entering the mesophyll cells, thereby slowing down photosynthesis and inhibiting the growth of vegetation.

Lagged correlations between vegetation growth and climate change do exist [26,27], as was also shown by this study. We comprehensively analyzed the response mechanism of NPP to climatic factors in the growing seasons and found that these factors played a decisive role in aspects such as the spatial distribution of vegetation types and the changes in vegetation productivity. Further investigation is needed regarding the underlying mechanism for large time lags between water use and vegetation activities in different climate conditions (e.g., arid, cold areas). Additionally, human activities and natural disturbances, e.g., agricultural irrigation, land-use change, forest development, fire, and plant diseases [28,29,30,31,32], could be other factors in vegetation growth. A comprehensive exploration of human and natural disturbances needs to be considered in subsequent research.

Although tree ring data were used to validate the regional vegetation NPP accuracy in this study, we know that tree ring archives could provide long-term information toward understanding regional changes in forest productivity and the carbon cycle under the influence of climate change. In our investigation, forest plots with fire disturbance and deadwood were also considered; therefore, further analysis of different kinds of plots (e.g., various altitudes and climatic and soil conditions) will be expanded to explore the influences of climate change and disturbances on vegetation growth.

## 4. Materials and Methods

### 4.1. Research Area

Northeast China is located within 38°40′~53°30′ N and 115°5′~132°2′ E (Figure 5). The region has a cold climate but a warm, short summer. It spans the temperate zone and the cold temperate zone. It has a monsoon climate, with precipitation occurring mainly from July to September. The annual precipitation increases from west to east, with the western region having an annual precipitation of only 250 to 400 mm, approximately half that of the eastern region. However, the solar radiation increases from north to south, with a longer duration in summer and a shorter duration in winter. The total radiation during the growing season accounts for 55 to 60% of that for the entire year, maintaining a frostless period of up to 130–170 days. The northeast region shows semi-arid, semi-humid, and humid temperature differences from west to east, whereas, in the north–south direction, the heat varies from that of a warm temperate zone to that of a temperate zone to a cold temperate zone with increasing latitude, forming a unique vegetation spatial distribution.

In this area, the major vegetation types on hills and mountains (Daxing’anling, Xiaoxing’anling, and the Changbai Mountain) are temperate broadleaved deciduous and needle-leaved evergreen forest. The major vegetation types in the western parts are semi-arid shrubs, grass, and temperate steppe. Croplands mainly distribute over Sanjing Plain, Songnen Plain, and Liaohe Plain.

### 4.2. Dataset

#### 4.2.1. Meteorological Data

The meteorological dataset employed in this study is the China Meteorological Forcing Dataset (CMFD), with a high spatial and temporal resolution, and which is published by the Cold and Arid Region Scientific Data Center. The spatial resolution of the dataset is 0.1°, which was achieved by integrating the meteorological observation data of the China Meteorological Administration and various other meteorological reanalysis data, mainly including reanalysis data from Princeton University in the USA, data products from the global land data assimilation system, the NASA Global Energy and Water Cycle Experiment (GEWEX) surface radiation budget dataset, as well as data acquired by tropical precipitation measurement satellites [33]. The temporal resolution was 3 h, including meteorological factors, such as near-surface temperature, air pressure, air humidity, wind speed, downward shortwave radiation, and precipitation. The CMFD data are considered highly reliable by many scholars [34,35], and the data used in this study were temperature, precipitation, air humidity, and solar radiation data acquired between 2003 and 2012. The daily maximum and minimum temperatures, daily precipitation, and solar radiation data were obtained after performing time scale and unit conversions. The time scale conversion was performed by taking the maximum and minimum values of the daily temperature as the highest and the lowest daily temperatures, respectively, while the sums of the daily precipitation and solar radiation were taken as the daily total precipitation and solar radiation data, respectively. The unit conversion was performed by converting the temperature unit to degrees Celsius (°C), the precipitation unit to millimeters (mm), and the solar radiation unit to megajoules per square meter (MJ/m^2^). The meteorological data were then resampled to ensure that the spatial resolution was the same as that of the leaf area index (*LAI*) data.

#### 4.2.2. Field Survey Data

The forest plot productivity survey was conducted between 5 and 23 August 2013, and between 9 and 24 August 2016 over Daxing’anling. As the northernmost ecological barrier and important forest conservation area in China, Daxing’anling contains the most representative boreal tree species. Therefore, forty-six plots over Daxing’anling were selected, including 17 deciduous broadleaf forests (*Betula platyphylla Suk*) plots, 14 deciduous needleleaf forests (*Larix gmelinii*) plots, 7 evergreen needleleaf forests (*Pinus sylvestris Linn.* var. *mongolica Litv.*) plots, and 8 mixed forest plots (Figure 5).

The plots were circular areas of land with a radius of 10 or 15 m. The surveyed vegetation types included major species in Daxing’anling, namely, *Larix gmelinii*, *Pinus sylvestris Linn. var. mongolica Litv.*, and *Betula platyphylla Suk*. Two aspects were investigated for each plot, namely, (1) the tree growth status in the plot and (2) the productivity of the forest. For every tree in the plot with a diameter at breast height (DBH) of more than 5 cm, a single tree measurement was performed to acquire the relevant data to establish a height–DBH model. By inputting the DBH obtained from the tree ring data into the growth models of different species, the height of a standard sample tree was derived. Tree ring data were acquired in the plot at six diameter levels (5–10 cm, 10–15 cm, 15–20 cm, 20–25 cm, 25–30 cm, and >30 cm). For each level, three standard tree samples were selected, of which, the tree cores were acquired by drilling at breast height (1.3 m) in both the east–west and the south–north directions to reduce measurement error without compromising the quality of the core [36]. A total of 596 tree cores was used for the tree ring analysis.

The results from the complete analysis of the tree cores are shown in Figure 6. Once a tree core was collected in the field and its length was measured, it was placed in a paper-sealed tube marked with the plot number and the DBH value and returned to the laboratory for processing. The tree cores were first air-dried and placed in a dry place for several weeks to stabilize. Subsequently, they were fixed on the wooden notch and polished with sandpaper until the tree rings became clearly visible. Lastly, the processed cores were scanned and measured to obtain the tree ring width and to calculate the annual growth. The tree ring width was measured using a special tree ring analysis instrument (WinDENDRO), with a measurement precision of 0.001 mm [37]. Due to its symmetry, the tree ring width could be measured in only one direction and, to eliminate interpretation error caused by a false ring during measurement, the tree rings in two directions of the tree were separately identified and plotted to obtain the framework diagrams (Figure 6). The tree ring widths were subsequently measured in both directions, the average of which was taken as the result. That is, tree ages and DBH increments were acquired through tree cores. The results were used to derive the annual growth of biomass using the forest growth model [38]. This model was adopted to derive the regional annual NPP, utilizing the conversion factors between carbon and biomass increments, and it was employed to verify the simulation results from the BEPS model.

#### 4.2.3. Remote Sensing Data

The remote sensing data used in this study included the GLASS *LAI* product dataset and the Moderate Resolution Imaging Spectroradiometer (MODIS) Land Cover Type product dataset (MCD12Q1) from the United States Geological Survey (USGS) acquired between 2003 and 2012. The *LAI* data were from the characteristic parameter product of GLASS. Based on the *LAI* products, such as MODIS and VEGETATION, as well as surface measurement data of global *LAI*, this product adopted the time series observation data integrated using a generalized neural network algorithm, making it suitable for long-term monitoring of vegetation growth on a regional scale [39]. The spatial resolution was 1 km, the temporal resolution was 8 d, and the time scale used in the study was from 2003 to 2012.

#### 4.2.4. Soil Texture Data

Soil texture data represent an important input parameter for the ecological process model. The BEPS model requires entering the percentage contents of sand, silt, and clay into the simulated grid. The soil texture dataset used in this study was obtained from the Land Surface Process and Resource Ecology Laboratory of the Institute of Global Change and Geosystem Science at Beijing Normal University. This dataset was produced based on the national 1:1,000,000-scale soil type map and the second national soil survey data. The values are expressed in percentages and the spatial resolution was 1 km.

### 4.3. BEPS Model

The BEPS model was developed from the FOREST-BGC ecological process model [10,40] to simulate vegetation processes, such as photosynthesis, respiration, carbon partitioning, and water and energy balance [13]. Therefore, the model includes several relevant modules, such as for photosynthesis, energy and water balance, and soil. During the simulation of vegetation productivity, the classic Farquhar photosynthesis model was adopted in BEPS to mimic the photosynthesis process of green plants [10]. However, the disadvantage of this simulation is that it only simulates the instantaneous photosynthesis of a single leaf; therefore, expansions on both the temporal and the spatial scales are required. The model realizes the conversion from the instantaneous scale to the daily scale using daily integration. As the various parts and tissues of the vegetation receive different quantities and quality of light, their photosynthesis intensities are also different. Therefore, treating the entire canopy as a large leaf will introduce substantial errors. Therefore, the BEPS model divides the canopy into sun leaves and shade leaves and introduces the leaf aggregation index to characterize the aggregation degree of the leaves [41]. This is intended to reduce the influence of the shape and structure of the plant canopy on radiation absorption, light received by the leaf’s surface, and mutual shading. By calculating the photosynthesis processes of sun and shade leaves separately, the BEPS model realizes the expansion of the canopy on the spatial scale, which eventually leads to the derivation of the gross primary production (*GPP*) of vegetation on a daily scale. The vegetation NPP is subsequently calculated by subtracting the autotrophic respiration of the vegetation from the *GPP*.

The model introduces the leaf aggregation index to calculate the leaf area index of the sun and shade leaves $LAI_{sun}$ and $LAI_{shade}$, respectively:(1)$$\begin{array}{l} LAI_{sun}=2\cos\theta (1-\exp(-0.5\Omega LAI/\cos\theta )),\\ LAI_{shade}=LAI-LAI_{sun} \end{array}$$
where LAI is the leaf area index of the plant canopy; θ is the sun zenith angle; Ω is the leaf aggregation index, which was defined based on different vegetation types.

The model subsequently calculates the solar radiation received by the sun and shade leaves, i.e., $S_{sun}$ and $S_{shade}$:(2)$$\begin{array}{l} S_{sun}=S_{dir}\cos\alpha /\cos\theta +S_{shade}\\ S_{shade}=(S_{dif}-S_{dif,under})/LAI+C\\ C=0.07\Omega S_{dir}(1.1-0.1LAI)\exp(-\cos\theta ),\\ S_{dif,under}=S_{dif}\exp(-0.5\Omega LAI/\cos\theta )\\ \cos\theta_{mean}=0.537+0.025LAI \end{array}$$
where $S_{dir}$ and $S_{dif}$ are the direct and scattered radiation received on the canopy, $S_{dif,under}$ is the scattered radiation received under the canopy, α is the average leaf solar radiation distribution angle, *C* is the multiple scattered radiation of the direct radiation, and $\theta_{mean}$ is the mean sun zenith angle of the scattered radiation.

The model subsequently calculates the total photosynthesis of the vegetation canopy:(3)$$A_{conopy}=A_{sun}LAI_{sun}+A_{shade}LAI_{shade},$$
where $A_{sun}$ and $A_{shade}$ are the photosynthesis levels of the sun and shade leaves, respectively.

Lastly, the model calculates the NPP:(4)$$\begin{array}{l} GPP=A_{conopy}L_{day}F_{GPP},\\ R_{a}=R_{m}+R_{g}=\sum \left(R_{m,i}+R_{g,i}\right) \end{array}$$
where *GPP* is the gross primary productivity; $L_{day}$ is the length of the day; $F_{GPP}$ is the proportional coefficient when converting from photosynthesis to *GPP*; $R_{a}$, $R_{m}$, and $R_{g}$ represent autotrophic respiration, maintenance respiration, and growth respiration, respectively; *i* represents the different parts of the vegetation (with 1, 2, and 3 representing the leaves, stems, and roots, respectively).

The input data of BEPS model contain daily meteorological factors (maximum and minimum temperature, precipitation, solar radiation, and relative humidity), land cover type, *LAI*, soil texture, latitude and longitude, and CO_2_ concentration.

### 4.4. Trend Analysis

Linear regression was used to analyze and establish the NPP trend, i.e., the relationship between the annual NPP and time. The calculation of the NPP trend can be expressed as:(5)y=a+kx+ε
where *y* is an estimated dependent variable (NPP); *x* is an independent variable (time); *k* is the regression coefficient of the linear equation, which is the quantified index of vegetation changes. The constants a and ε are the constant and error terms, respectively. The method of least absolute deviation was used to calculate the constants of the linear regression model, with the criterion of the sum of the absolute deviations of the observed value and the estimated value being the smallest [42].

### 4.5. Partial Correlation Analysis

To eliminate the influence of other variables, a partial correlation was adopted to investigate the independent correlation of the vegetation NPP with temperature, precipitation, solar radiation, and relative humidity. *R* was used to represent the correlation coefficient, with the calculation formula as follows:(6)$$R_{1234}=\frac{R_{123}-R_{143}R_{243}}{\sqrt{\left(1-R_{143}^{2})(1-R_{243}^{2}\right)}},$$
where variable 1 denotes the vegetation NPP, and variables 2, 3, and 4 denote different meteorological factors. $R_{1234}$ is the partial correlation coefficient between variables 1 and 2 when variables 3 and 4 are fixed. $R_{123}=\frac{R_{12}-R_{13}R_{23}}{\sqrt{\left(1-R_{13}^{2})(1-R_{23}^{2}\right)}}$, while the calculations of $R_{143}$ and $R_{243}$ are the same as that of $R_{123}$, $R_{12}$, $R_{13}$, and $R_{23}$, which are the simple linear correlation coefficients between variables 1 and 2, 1 and 3, and 2 and 3, respectively.

### 4.6. Lag Analysis

Whereas part of the precipitation is absorbed by vegetation, the remaining precipitation either becomes surface runoff or is preserved in the soil after permeation, both of which still constantly contribute to vegetation growth. Therefore, only analyzing the effect of a fixed period of precipitation on vegetation productivity cannot accurately reflect the relationship between the two. To solve this issue, this study adopted lag analysis to quantify the lag in the response of vegetation growth to changes in precipitation.

Previous studies generally found the time lag at the monthly scale of vegetation response to climate to be short [43,44,45]. Therefore, we selected the cumulative lag month and time lag to quantitatively measure the lag in the response of the vegetation NPP to precipitation change. Subsequently, we calculated the correlation coefficient between the two factors. Starting from the vegetation NPP in a certain month, the number of previous months when the cumulative precipitation value was recorded increased successively, which was defined as the cumulative lag month. In this study, as the water absorbed during vegetation growth not only included the precipitation from the current month but also the previous several months, the delay in the month was defined as the time lag (lag*x*).

Vegetation growth in Northeast China mainly occurs from May to September (growth season), when the time lags were also the most significant. This study calculated the lagged correlation coefficient between the vegetation NPP and precipitation in the growing seasons. The month of July is used below as an example to describe the specific steps of the lag analysis:(1)Calculate the correlation coefficient between the vegetation NPP and precipitation in July between 2003 and 2012, i.e., for a lag period of 0 months (lag0) at a cumulative number of lag months of 1.(2)Calculate the correlation coefficient between the vegetation NPP in July and cumulative precipitation in June and July between 2003 and 2012, i.e., for a lag period of 0 months (lag0) and a cumulative number of lag months of 2.(3)On the same basis, calculate the correlation coefficients of the vegetation NPP for a lag period of 0 months and the cumulative precipitation of 3, 4, and 5 months.(4)Calculate the correlation coefficient between the vegetation NPP in July and precipitation in June, i.e., for a lag period of 1 month (lag 1) and a cumulative number of lag months of 1.(5)Calculate the correlation coefficient between the vegetation NPP in July and the cumulative precipitation in May and June, i.e., for a lag period of 1 month (lag 1) and a cumulative number of lag months of 2.(6)On the same basis, calculate the correlation coefficients between the vegetation NPP in July at a lag period of 1 month and the cumulative precipitation of 3, 4, and 5 months.(7)On the same basis, calculate the correlation coefficients of the vegetation NPP in July for different lag periods with different cumulative numbers of lag months.

### 4.7. Evaluation and Analysis of the Modeled Estimates

To evaluate the simulated NPP, we used the results derived from the tree ring data as ground truth observations. We calculated determination coefficient (*R*^2^) (Equation (7)) and the root mean square error (*RMSE*) (Equation (8)) to evaluate the accuracy of the BEPS model simulation. Additionally, a significance test (*p*-value) was conducted to disprove the concept of “chance” and to reject a null hypothesis by adhering to the observed patterns.
(7)$$R^{2}=1-\frac{\sum_{i=1}^{t}{\left(X_{obs}-X_{\mathrm{mod}}\right)}^{2}}{\sum_{i=1}^{t}{\left(X_{obs}-\bar{X_{\mathrm{mod}}}\right)}^{2}},$$
(8)$$RMSE=\sqrt{\frac{\sum_{i-1}^{t}{\left|X_{obs}-X_{\mathrm{mod}}\right|}_{i}^{2}}{t}},$$

## 5. Conclusions

In this study, based on meteorological, forest survey, soil texture, remote sensing, and other auxiliary data, we simulated the vegetation NPP in Northeast China between 2003 and 2012 with the BEPS model and verified it with tree ring data. The results indicated the simulation to be accurate and the tree ring data to be a reliable means for verifying the time series vegetation productivity on a regional scale, with *R*^2^ = 0.84 and *RMSE* = 42.73 gC/m^2^·a.

The annual NPP over Northeast China showed evident spatial gradients, where they were generally higher in the northeast than in the southwest part of the study area, which could be explained by spatial variabilities in the vegetation types and climate. During the study period, the annual NPP showed a significant increasing trend over Northeast China, particularly in forest lands, whereas the areas showing a reducing trend were mainly in farmlands and grasslands.

Partial correlation analysis suggested that the NPP was more sensitive to temperature and solar radiation. Further analysis of the time lag in the response of the NPP to precipitation changes suggested that the lag period of the forests was substantially longer than that of the grasslands and farmlands. This was because forests have a developed root system and, consequently, present a stronger soil and water conservation function than other vegetation types do. Alternatively, vegetation growth in grasslands and farmlands not only relies on the precipitation of the current month but also on the soil moisture conditions. Since soil moisture is the result of cumulative precipitation over multiple months, grasslands and farmlands have a longer cumulative number of lag months than forests do. The authors are confident that the results of this study can provide scientific references for vegetation management under changing climate conditions.

## Figures and Tables

**Figure 1 plants-10-00821-f001:**
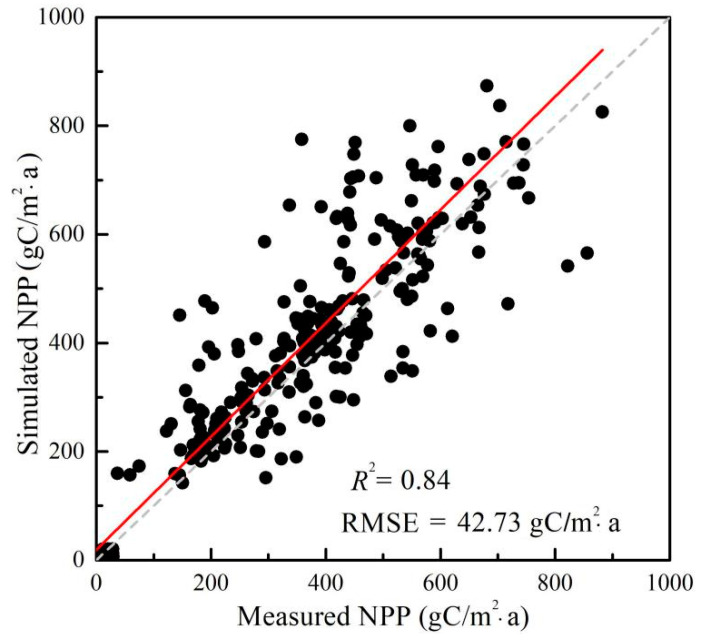
Validation of the simulated net primary productivity (NPP) from the boreal ecosystem productivity simulator (BEPS) model.

**Figure 2 plants-10-00821-f002:**
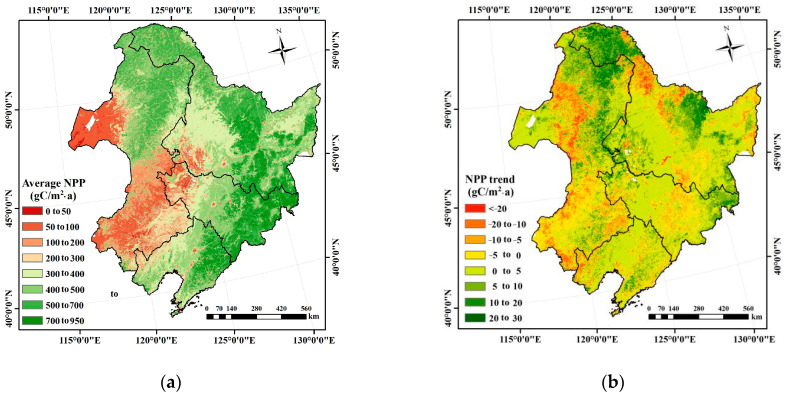
Spatial and temporal variation characteristics of the vegetation NPP in Northeast China: (**a**) spatial distribution of the average NPP over multiple years and (**b**) the NPP trend.

**Figure 3 plants-10-00821-f003:**
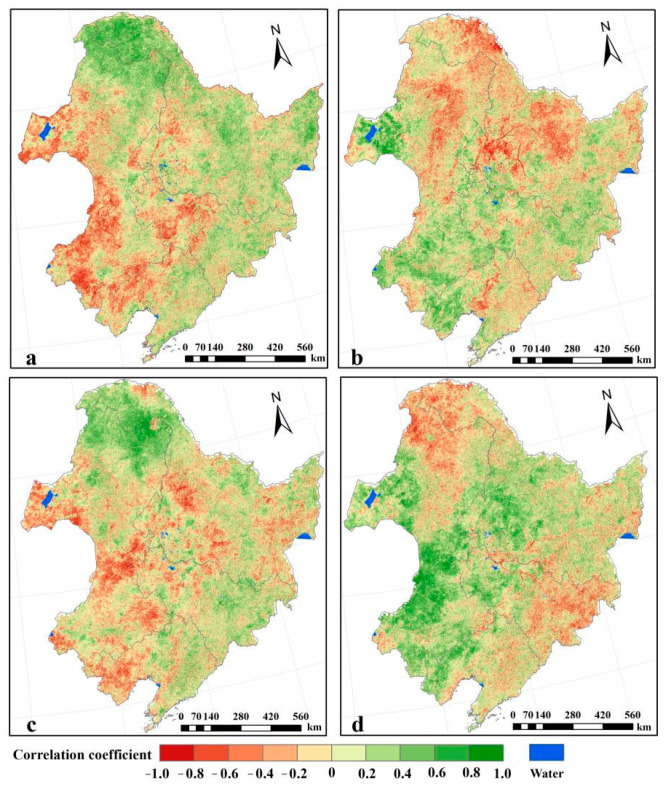
Results of the partial correlation analysis between the vegetation NPP and meteorological factors during 2003 to 2012: (**a**) temperature, (**b**) precipitation, (**c**) solar radiation, and (**d**) relative humidity.

**Figure 4 plants-10-00821-f004:**
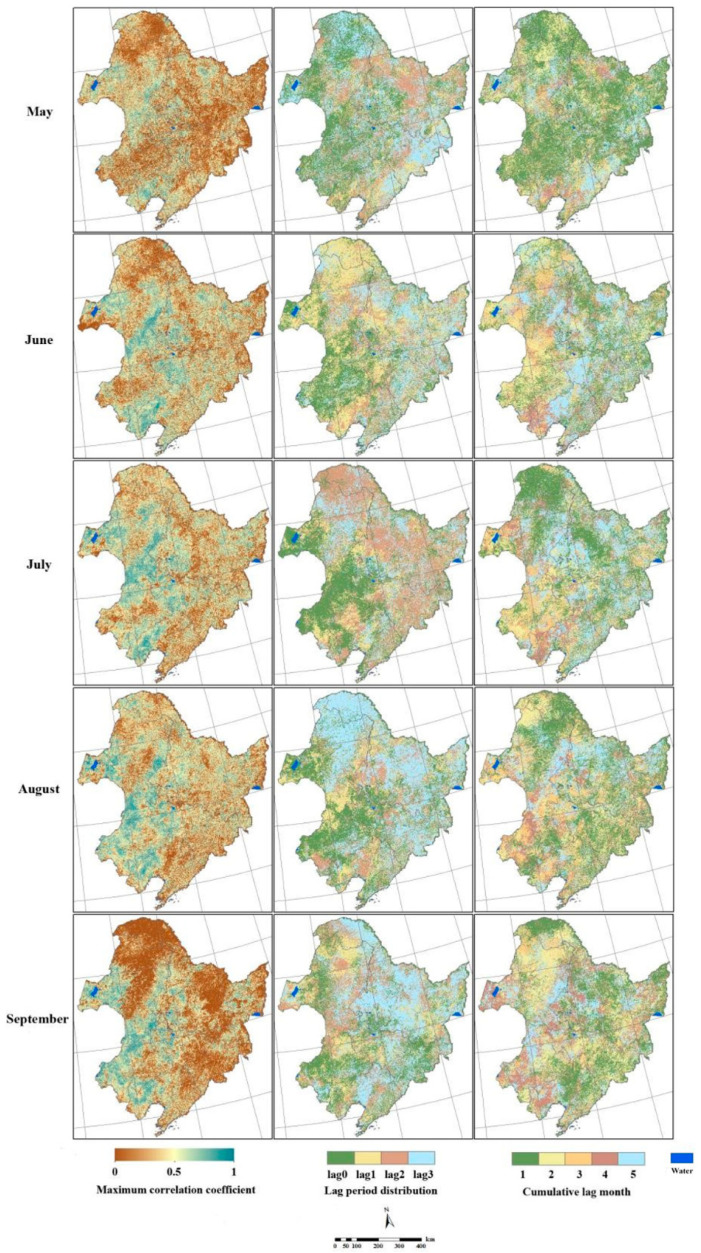
Lag responses of the NPP to the precipitation over Northeast China.

**Figure 5 plants-10-00821-f005:**
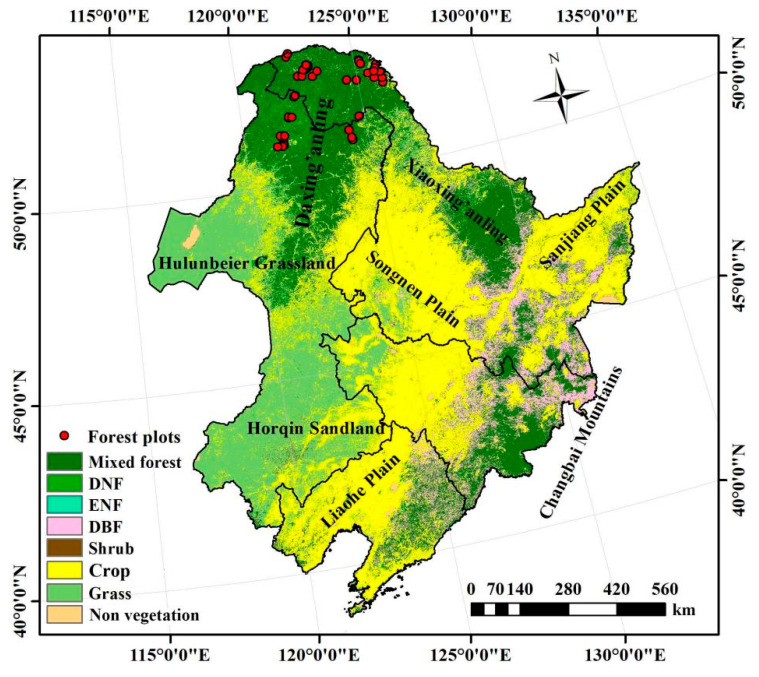
Distribution of the vegetation types and forest plots in Northeast China (DNF: deciduous needleleaf forest, ENF: evergreen needleleaf forest, DBF: deciduous broadleaf forest).

**Figure 6 plants-10-00821-f006:**
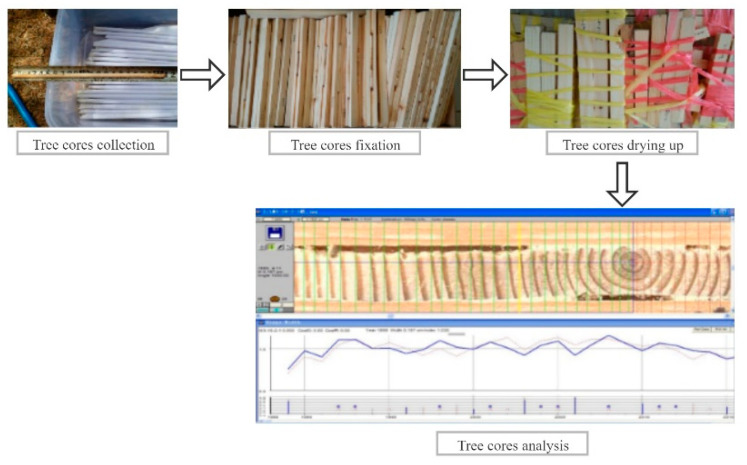
Flowchart of tree rings analysis.

## Data Availability

Not available.

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
