# Peer review of "A Decade’s Change in Vegetation Productivity and Its Response to Climate Change over Northeast China"

_plants, 2021, doi:10.3390/plants10050821_

Round 1
Reviewer 1 Report
It is a hot topic to examine the effects of global climate change on the boreal vegetation in Northeast China. Overall, the manuscript is well written and organized.
Main comments:
L32 which was mainly ascribed to.
L33-34 Vegetation is an important carbon sink and source, storing large amounts of carbon in plant parts and the soil. It would be better to change "vegetation" to "terrestrial ecosystem".
L35 Change "better" to "well".
L60 Both studies suggested that climate change would increase the vegetation NPP. It means the past climate change or future predictions? Please clarify.
The unit of NPP (Figure 2, 3) should be consistent. There is a space between g and C.
Author Response
Dear editor and reviewer,
Thank you for your constructive suggestions, and here are the responses point to point.
Sincerely
Min
Comments and Suggestions for Authors
It is a hot topic to examine the effects of global climate change on the boreal vegetation in Northeast China. Overall, the manuscript is well written and organized.
Main comments:
- L32 which was mainly ascribed to.
Answer: Thank you for your suggestion. We have revised this sentence in the revised manuscript. See line 32.
- L33-34 Vegetation is an important carbon sink and source, storing large amounts of carbon in plant parts and the soil. It would be better to change "vegetation" to "terrestrial ecosystem".
Answer: Thank you for your suggestion. We have changed "vegetation" to "terrestrial ecosystem" in the revised manuscript. See line 33.
- L35 Change "better" to "well".
Answer: Thank you for your suggestion. We have revised "better" to "well" in the revised manuscript. See line 36.
- L60 Both studies suggested that climate change would increase the vegetation NPP. It means the past climate change or future predictions? Please clarify.
Answer: Thank you for your suggestion. This sentence mean that the previous studies listed in the manuscript (reference [15, 16]) proved this fact: “The combined effects of climate change and CO2 fertilization on the increase of NPP were estimated to be 10-12% for 2030s and 28-37% in 2090s.” See line 67.
We have revised the sentence to make it more understandable.
- The unit of NPP (Figure 2, 3) should be consistent. There is a space between g and C.
Answer: Thank you for your suggestion. We have revised the figures and all the units in the manuscript.

Reviewer 2 Report
Dear Authors,
I am sending a review of the article “Decade change of vegetation productivity and its response to climate change over Northeast China”.
Comments
Line 75-85 – The annual precipitation increases from west to east, with the western region having an annual precipitation of only 250 mm to 400 mm, approximately half of that of the eastern region. However, the solar radiation increases from north to south, with a longer duration in summer and a shorter duration in winter. The total radiation during the growing season accounts for 55% to 60% of that for the entire year, maintaining a frostless period of up to 130–170 days. The northeast region shows semi-arid, semi-humid, and humid temperature differences from west to east, whereas, in the north–south direction, the heat varies from that of a warm temperate zone, temperate zone to cold temperate zone with increasing latitude, forming a unique vegetation spatial distribution.
Rev – vegetation and climate are very different. Why such surfaces for field study were chosen? The choice must be explained in the text. The vegetation is representative? The tree species are representative? The tree species vere analysed together or separately? The text should be completed.
Line 119 – 2.2.2. Field survey data are insufficient.
Rev – to evaluate the simulated NPP, Authors used the results derived from tree ring data as ground observations. In my opinion the section 2.2.2. Field survey data are insufficient.
Line 124 – … The plots were circular lands with a radius of 10 or 15 cm.
Rev – ??? what these surfaces were in reality? In what type of vegetation the research was conducted? Mixed forest or Deciduous needleleaf forest?
Line 124-125 – … The surveyed vegetation types mainly included major species, such as Xing’an larch and white birch.
Rev – this information is very modest.
Line 127 – … For every tree in the plot with a diameter …
Rev – that is, the Xing’an larch and white birch were studied?
Rev – how many trees were studied?
Rev – whether altitude was taken into account?
Rev – how many trees were in each plot?
Rev – English and latin name of studied tree species should be given.
Line 141-142 – … processed cores were scanned and measured to obtain the tree ring width and to calculate the annual growth.
Rev – addition a sample scan (aditional figure) and measurement results (aditional table or figure) can significantly increase the attractiveness and value of the article.
Line 142-145 – … The results were used to derive the annual growth of biomass using the forest growth model. This model was adopted to derive the regional annual NPP utilizing the conversion factor between carbon and biomass increment, and it was employed to verify the simulation results from the BEPS model … .
Rev – this sentence should be placed at the end of the section 2.2.2. Field survey data.
Line 348-349 – … Figure 4. Results of partial correlation analysis between vegetation NPP and meteorological factors (a: temperature; b: precipitation; c: solar radiation; d: relative humidity) … .
Rev – information about the research period should be added.
Technical notes
Line 31 – … in 2017[1], … . Line 55 – … change [15], … .
Rev – spaces should be unified throughout the all manuscript.
Line 217 – … and CO2 concentration … .
Rev – correction required.
Rev – Literature should be prepared according to the journal's requirements.
Author Response
Dear editor and reviewer,
Thank you for your constructive suggestions, and here are the responses point to point.
Sincerely
Min
Comments
- Line 75-85 – The annual precipitation increases from west to east, with the western region having an annual precipitation of only 250 mm to 400 mm, approximately half of that of the eastern region. However, the solar radiation increases from north to south, with a longer duration in summer and a shorter duration in winter. The total radiation during the growing season accounts for 55% to 60% of that for the entire year, maintaining a frostless period of up to 130-170 days. The northeast region shows semi-arid, semi-humid, and humid temperature differences from west to east, whereas, in the north-south direction, the heat varies from that of a warm temperate zone, temperate zone to cold temperate zone with increasing latitude, forming a unique vegetation spatial distribution.
Rev – vegetation and climate are very different. ① Why such surfaces for field study were chosen? The choice must be explained in the text. The vegetation is representative? The tree species are representative? ② The tree species were analysed together or separately? The text should be completed.
Answer: Thank you for your constructive suggestion.① Forests in northeast China mainly distribute in Daxing’anling, Xiaoxing’anling, and Changbai Mountain. Daxing’anling is the north most ecological barrier and important forest conservation area in China. It belongs to representative temperate and continental climate. The major species in Daxinganlign include Larix gmelinii, Pinus sylvestris Linn. var. mongolica Litv., and Betula platyphylla Suk. In our field survey, forty-6 forest plots contain 17 Betula platyphylla Suk plots,14 Larix gmelinii plots, 7 Pinus sylvestris Linn. var. mongolica Litv. plots, and 8 mixed forest plots. We selected representative forest plots within few step: first, we viewed the forest distribution and terrain conditions over Daxing’anling using high-resolution images and DEM; then considering the natural and dense forest over Daxing’anling, we selected some forest plots on Google earth image randomly along forest paths where we could reach; finally, we adjusted the random forest plots in the real field work which involved in different species.
② Tree rings of different species were analysed separately, and then the DBH was obtained. Different growth equations of different species were used to calculate the aboveground biomass.
Related information was added in the revised manuscript.
- Line 119 – 2.2.2. Field survey data are insufficient.
Rev – to evaluate the simulated NPP, Authors used the results derived from tree ring data as ground observations. In my opinion the section 2.2.2. Field survey data are insufficient.
Answer: Thank you for your constructive suggestion. We have added some related information in this section.
- Line 124 – … The plots were circular lands with a radius of 10 or 15 cm.
Rev – ??? what these surfaces were in reality? In what type of vegetation the research was conducted? Mixed forest or Deciduous needleleaf forest?
Answer: Thank you for your constructive suggestion. Among the forty-six forest plots, 14 deciduous needleleaf forest (Larix gmelinii) plots, 17 deciduous broadleaf forest (Betula platyphylla Suk) plots, 7 evergreen needleleaf forest (Pinus sylvestris Linn. var. mongolica Litv.) plots, and 8 mixed forest plots were included. Pictures about Larix gmelinii and Pinus sylvestris Linn. var. mongolica Litv. Plots are shown in Figure1.
Figure 1 Forest plots over Daxing’anling
Left: Larix gmelinii and Betula platyphylla; Right: Suk mixed plot Pinus sylvestris Linn. var. mongolica Litv.
- Line 124-125 – … The surveyed vegetation types mainly included major species, such as Xing’an larch and white birch.
Rev – this information is very modest.
Answer: Thank you for your constructive suggestion. We have revised this section overall together with question 1, 2 and 3.
- Line 127 – … For every tree in the plot with a diameter …
① Rev – that is, the Xing’an larch and white birch were studied?
Answer: Thank you for your constructive suggestion. Exactly, Xing’an larch (Larix gmelinii) and white birch (Betula platyphylla Suk) were all considered in the field investigation. Each tree in with a diameter greater than 5 cm was cored in the forest plot. See section 4.2.2.
② Rev – how many trees were studied?
Answer: Thank you for your constructive suggestion. Totally, 46 forest plots and 596 trees were studied. Tree rings at east-west and south-north directions of each tree were required in the field work. See line 297 in the revised manuscript.
③ Rev – whether altitude was taken into account?
Answer: Thank you for your constructive suggestion. Meteorological data employed in this study was from CMFD, which introduced high-resolution elevation data in the interpolation of air temperature and pressure, as both of them are sensitive to altitude. Therefore, topographic Factors are not involved in the model simulation as independent input data.
④ Rev – how many trees were in each plot?
Answer: Thank you for your constructive suggestion. The number of trees depends on the real conditions of each forest plot. The specific numbers ares listed in figure 2.
Figure 2. Number of trees of the 46 forest plots
⑤ Rev – English and latin name of studied tree species should be given.
Answer: Thank you for your constructive suggestions. The names of the tree species are revised in the manuscript.
- Line 141-142 – … processed cores were scanned and measured to obtain the tree ring width and to calculate the annual growth.
Rev – addition a sample scan (additional figure) and measurement results (additional table or figure) can significantly increase the attractiveness and value of the article.
Answer: Thank you for your constructive suggestion. When we took the tree rings back to the library, the tree cores were first fixed on the sticks with notches (Figure 3). The tree cores were then air-dried and placed in a dry place for several weeks. Tree cores polishing with sandpaper is a crucial step to make sure the tree rings visible and readable. All the polished tree cores were scanned and analyzed using WinDENDRO software (Figure 4). Meanwhile, the “ring width” window was showed in the software. Tree ring width were measured in both directions to grantee the accuracy of the width and recognize the false tree rings.
The related information was added in the revised manuscript. See section 4.2.2.
Figure 3 Tree cores were fixed to the notches
Figure 4 The scanned tree core in WinDENDRO software and “rings width” window
- Line 142-145 – … The results were used to derive the annual growth of biomass using the forest growth model. This model was adopted to derive the regional annual NPP utilizing the conversion factor between carbon and biomass increment, and it was employed to verify the simulation results from the BEPS model.
Rev – this sentence should be placed at the end of the section 2.2.2. Field survey data.
Answer: Thank you for your constructive suggestions. These sentences were moved at the end of the section 4.2.2, which were highlighted in the revised manuscript.
- Line 348-349 – … Figure 4. Results of partial correlation analysis between vegetation NPP and meteorological factors (a: temperature; b: precipitation; c: solar radiation; d: relative humidity) … .
Rev – information about the research period should be added.
Answer: Thank you for your constructive suggestion. The research period was added in the revised manuscript, See line 141-143.
Technical notes
- Line 31 – … in 2017[1], … . Line 55 – … change [15], … .
Rev – spaces should be unified throughout the all manuscript.
Answer: Thank you for your constructive suggestion. All the format issues were checked in the revised manuscript.
- Line 217 – … and CO2 concentration … .
① Rev – correction required.
Answer: Thank you for your constructive suggestion. This sentence was corrected, which was highlighted in the revised manuscript. See line 385.
② Rev – Literature should be prepared according to the journal's requirements.
Answer: Thank you for your constructive suggestion. References were corrected according to the journal’s requirements. See Reference section.

Reviewer 3 Report
This study is of great interest worldwide, but few changes and corrections are required before proceeding it to further steps. Brief introduction of the climate change as well as vegetation types is required in the introduction section. Proper placement of few sections is required like why “Research Area and Dataset” has been placed as a separate section? It should be merged in M & M section. The methodology mentioned in “2.2.2. Field survey data” are novel? First time used and reported by authors?? Obviously no!!, so, cite most parts with the relevant citations. First sentence of this section needs rephrasing “The forest plot productivity survey was conducted from 5–23 August 2013, and again from 9–24 August 2016”. Please check the unit, is this correct “The plots were circular lands with a radius of 10 or 15 cm”? Methods section should be changed to Statistical work/ analysis.
The authors didn’t clearly mention the dendrochronological part in results and discussion sections that needs a brief elaboration. Proper source is required for Figures: How was this map generated? Mention the name of software as well as the version.
As a final note, let me say that I sincerely hope that the author will find these suggestions helpful.
Author Response
Dear editor and reviewer,
Thank you for your constructive suggestions, and here are the responses point to point.
Sincerely
Min
Comments
Comments and Suggestions for Authors
This study is of great interest worldwide, but few changes and corrections are required before proceeding it to further steps.
- Brief introduction of the climate change as well as vegetation types is required in the introduction section.
Answer: Thank you for your constructive suggestion. We have added related information in introduction, and more information about tree species were added in 4.2.2 section.
- Proper placement of few sections is required like why “Research ” has been placed as a separate section? It should be merged in M & M section.
Answer: Thank you for your constructive suggestion. “Research Area and Dataset” was moved to M&M section, and the whole structure of the manuscript was adjusted according to the journal’s requirements.
- The methodology mentioned in “2.2.2. Field survey data” are novel? First time used and reported by authors?? Obviously no!!, so, cite most parts with the relevant citations. First sentence of this section needs rephrasing “The forest plot productivity survey was conducted from 5–23 August 2013, and again from 9–24 August 2016”. Please check the unit, is this correct “The plots were circular lands with a radius of 10 or 15 cm”? Methods section should be changed to Statistical work/ analysis.
Answer: Thank you for your constructive suggestion. The citations were added in section 4.2.2. The first sentence of this section was corrected in the revised manuscript. All the units of the manuscript were checked and revised. Actually, the radius of the forest plot depends on the real conditions, and all of them are 10 or 15 cm.
Methods section in the original manuscript was adjusted according to the journal’s requirements, and the statistical work/analysis as a part of the methods was included in the M&M section in the revised manuscript.
- The authors didn’t clearly mention the dendrochronological part in results and discussion sections that needs a brief elaboration. Proper source is required for Figures: How was this map generated? Mention the name of software as well as the version.
Answer: Thank you for your constructive suggestion. The dendrochronological part was expanded in 4.2.2 section in the revised manuscript. Additionally, we have added the information in discussion part. Figure 1 was generated in Origin 8.0, and figure 2 to 5 was mapped in Arcmap 10.5.1.
As a final note, let me say that I sincerely hope that the author will find these suggestions helpful.

Round 2
Reviewer 2 Report
Dear Authors,
I am sending a review of the article “Decade change in vegetation productivity and its response to climate change over Northeast China”.
Note
Rev – literature needs to be corrected
Zhao, M.; Running, S.W. Drought-Induced Reduction in Global Terrestrial Net Primary Production from 2000 Through. Science 2010, 329, 940–943, doi:10.1126/science.1192666.
Rev – the same journal – caps in title
Beer, C.; Reichstein, M.; Tomelleri, E.; Ciais, P.; Jung, M.; Carvalhais, N.; Rödenbeck, C.; Arain, M.A.; Baldocchi, D.; Bonan, G.B.; et al. Terrestrial gross carbon dioxide uptake: Global distribution and covariation with climate. Science 2010, 329, 834–838, 496 doi:10.1126/science.1184984.
Rev – the same journal – no all caps
Rev – the note applies to all literature
References should be described as follows, depending on the type of work (https://www.mdpi.com/journal/plants/instructions).